# Pollutant Absorption as a Possible End-Of-Life Solution for Polyphenolic Polymers

**DOI:** 10.3390/polym11050911

**Published:** 2019-05-20

**Authors:** Thomas Sepperer, Jonas Neubauer, Jonas Eckardt, Thomas Schnabel, Alexander Petutschnigg, Gianluca Tondi

**Affiliations:** Forest Products Technology & Timber Constructions Department, Salzburg University of Applied Sciences, Marktstraße 136a, 5431 Kuchl, Austria; thomas.sepperer@fh-salzburg.ac.at (T.S.); jonas.neubauer@fh-salzburg.ac.at (J.N.); jonas.eckardt@fh-salzburg.ac.at (J.E.); thomas.schnabel@fh-salzburg.ac.at (T.S.); alexander.petutschnigg@fh-salzburg.ac.at (A.P.)

**Keywords:** Bio-based foams, wastewater treatments, cationic dyes, anionic surfactants, pollutant adsorbents, tannin polymer, tannin-furanic foam

## Abstract

Tannin- and lignin-furanic foams are natural porous materials that have attracted high interest in the scientific and industrial communities for their high thermal and fire-resistant properties. However, no interesting solutions have been proposed for the management of their end-life as yet. In this study, the phenolic-furanic powders derived from the foams were analyzed for their capacity to remove different pollutants like neutral, cationic, and anionic organic molecules from wastewater. It was observed that the macromolecules produced from initially bigger fractions were more suitable to remove methylene blue and sodium dodecyl sulfate (SDS) while contained absorptions were observed for riboflavin. Acidified tannin powders were also prepared to understand the role of the flavonoid in the absorption mechanism. The latter showed outstanding absorption capacity against all of the tested pollutants, highlighting the key-role of the flavonoid fraction and suggesting the limited contribution of the furanic part. All adsorbents were investigated through FT-IR and solid state ^13^C-NMR. Finally, the powders were successfully regenerated by simple ethanol washing, showing almost complete absorption recovery.

## 1. Introduction

Water plays a major role in many industrial activities such as the textile dyeing and washing, chemical manufacturing, and heavy-metal mining [1,2,3,4,5]. During these processes, the water is often contaminated with different pollutants including, but are not limited to, surfactants (e.g., alkylphenol ethoxylates, sodium dodecyl sulfate), heavy metal ions (e.g., lead, copper, mercury), and dyes (methylene blue, amaranth or tartrazine). Therefore, effluents derived from these industries have to be regenerated before the water is reinserted into its natural cycle.

There are several strategies to approach wastewater treatment that range from well-known processes like filtration, flocculation, and decantation to specially designed adsorbents or by-products of the agricultural industry [6,7,8,9].

A recent study presented interesting organic hydrogels and inorganic nanocomposites for the removal of methylene blue dyes with very high adsorptions [10,11,12]. Anionic surfactants are generally removed from wastewater by using anaerobic bacteria [13,14], though cationic polymers have recently also been applied as absorbents [15]. Neutral organic molecules are removed by using active carbons [16,17].

Natural polyphenols such as lignin and tannins are characterized by high numbers of aromatic and hydroxyl groups [18] and their combination with furanic structures [19] could further enhance their capacity to establish tight secondary forces with organic pollutants.

First generation tannin-furanic foams have already been studied for their capacity to absorb pollutants [20,21,22,23] and heavy-metals [24,25], but unfortunately contained formaldehyde, which have hindered their development as an insulation material for buildings. 

More recently, formaldehyde-free furanic foams made of tannin and lignin have been prepared [26,27]. These products have attracted high scientific and industrial interest due to their completely natural skeletal constitution, their high thermal insulating performance, and their fire and water resistance [28,29,30]. At present, the major drawbacks for these materials are: (i) the heterogeneous nature of the tannin extract that contains several non-phenolics like sugars and proteins, and (ii) the undefined strategy for the end-life of this material. In recent studies, easy extraction processes for improving the purity of the fraction have been carried out [31,32] and tannin foams based on these extract have also been synthesized [33]. However, regarding the end-life of the product, no alternatives to the partial recycling of the foams have been proposed as yet [34]. 

The objective of this study was to observe the absorption properties of the polyphenolic-furanic powders derived from the foams against three different classes of pollutants (anionic, cationic, and neutral) in order to find an interesting end-life application for these materials. 

## 2. Materials and Methods

### 2.1. Chemicals

Mimosa (Acacia mearnsii) tannin powder “Weibull AQ” was supplied by Tanac SA (Montenegro, RS, Brazil). Black liquor was supplied by Lenzing AG (Lenzing, Austria). For the production of the foams, furfuryl alcohol was supplied by Trans Furans Chemicals (Geel, Belgium). Diethylether and sulfuric acid 98% were purchased from Merck (Darmstadt, Germany). The other chemicals used were technical grade acetone, formaldehyde (37%), chloroform, methylene blue, sodium dodecyl sulfate and riboflavin, all purchased from VWR (Darmstadt, Germany) and Roth (Karlsruhe, Germany).

### 2.2. Adsorbent Preparation

The preparation of the foams was undertaken by homogenizing the tannin (or tannin-black liquor (BL) mixture) with water, furfuryl alcohol (FOH), formaldehyde (37% water solution), and diethylether (DEE). Afterward, sulfuric acid (4 M) was added and the suspension was further stirred to obtain a homogeneous blend that was poured into a wooden mold. The mold was then covered and placed in a laboratory hot press for 5 min at different temperatures for expansion. Table 1 summarizes the produced foams and the exact synthesis procedure can be found in previous studies [27,29,33,34].

The foams were successively pulverized using a mortar and pestle and washed with distilled water. Afterward, the powders were dried in a convection oven at 103 °C and finally washed with ethanol to remove all unreacted chemicals. After drying, the powders were sieved and those of a particle size up to 250 µm were used for the adsorption experiments.

Acidified tannin gel was prepared following the procedure described by [5,35,36] with modifications concerning the acid amount and reaction time. In more detail, the tannin gel was synthesized by mixing 20% of tannin powder with 40% water, afterward, 40% of concentrated sulfuric acid (98%) was added and a reflux condenser was attached to the flask. The mixture was kept under magnetic stirring and heated to 100 °C for 2 h. The precipitate was recovered through glass filtering and this acidified tannin gel was used as a flavonoid reference after washing with water and ethanol and final drying at 103 °C.

### 2.3. Adsorption Experiments

Three different water contaminants were tested: (i) riboflavin, (ii) sodium dodecyl sulfate, and (iii) methylene blue (structures shown in Figure 1). Methylene blue was chosen to represent cationic dyes, particularly since this compound can be hazardous in higher concentrations [7]. The second class of chemicals were anionic surfactants and were represented by sodium dodecyl sulfate, which is harmful to water organisms as it denatures proteins and also causes irritation to humans [37,38]. As a representative for neutral charged components, but also to represent pharmaceuticals, riboflavin was the third tested pollutant. Aqueous solutions containing 20 ppm of the pollutants were prepared to determine the absorption capacity. An equivalent of 1 mg of absorbent was added per 5 mL of the solution and magnetically stirred for 48 h in the dark to avoid photo-degradation of the compounds at ambient temperature. After this period, the adsorption was considered complete and saturation of the adsorbents was expected [21].

The remaining concentration of riboflavin and methylene blue were determined using a Shimadzu UVmini 1240 UV/vis spectrophotometer (Shimadzu, Kyoto, Japan). After adsorption, the solutions were centrifuged at 3000 rpm for 3 min and the absorption of the supernatant was measured at 450 nm for riboflavin and 656 nm for methylene blue. A multipoint calibration curve for both compounds was created (*R*^2^ = 0.9999).

To determine the remaining SDS concentration, the assay described by [39] was used. Therefore, a 0.5 % methylene blue solution was diluted 100 fold with a pH stable medium (pH 7.2). A total of 0.5 mL of the sample (after adsorption) was mixed with 0.25 mL of the methylene blue solution in a reaction vial. Afterward, 1.5 mL of chloroform was added and vortexed twice for 3 s each. The mixture was then centrifuged at 3000 rpm for 3 min. Then, the chloroform layer was transferred to a quartz cuvette and the absorption was read at 656 nm. All measurements were performed in triplicate.

Furthermore, the adsorption capacity *q* of each foam was calculated using Equation (1). The results are expressed as mg pollutant absorbance per g adsorbent.
(1)$$q=\frac{\left(C_{0}-C_{1}\right)\cdot V}{W}$$
where *C*_0_ and *C*_1_ are the initial and the equilibrium concentration of pollutant (ppm), respectively, and *V* (mL) and *W* (mg) are the volume of the solution and the mass of the adsorbent, respectively.

### 2.4. ATR FT-IR Investigation

The polyphenolic powders (tannin foams, acidified tannin gel, and original tannin powder) were scanned with an FT-IR Frontier (Perkin-Elmer, Waltham, MA, USA) spectrometer coupled with an ATR miracle unit. The spectra were registered in triplicate with 32 scans in the spectral region between 4000 and 600 cm^−1^ with a resolution of 4 cm^−1^. The spectra were then investigated in the spectral region between 1800 and 600 cm^−1^, averaged, normalized, and baseline corrected with the software KnowItAll (BioRad, California, USA).

### 2.5. Solid State ^13^C-NMR

Spectra of the standard tannin foam, the acidified tannin gel, and the original tannin powder were obtained on a Bruker Avance NEO 500 wide bore system (Bruker BioSpin, Rheinstetten, Germany) at the NMR Center of the Faculty of Chemistry, University of Vienna. A 4 mm triple resonance magic angle spinning (MAS) probe was used with a resonance frequency for ^13^C at 125.78 MHz and the MAS rotor spinning was set to 14 kHz. Cross polarization (CP) was achieved by a ramped contact pulse with a contact time of 2 ms. During acquisition, ^1^H was high power decoupled using SPINAL with 64 phase permutations. The chemical shifts for ^13^C are reported in ppm and are referenced external to adamantane by setting the low field signal to 38.48 ppm.

The data elaboration was done with the software Top-spin 4.0.6 (Bruker) and OriginPro (OriginLab) while the calculations of the theoretical chemical shifts were done with the software NMR-Predict developed by the University of Lausanne (L. Patiny) and the University of del Valle (J. Wist) [40,41,42].

### 2.6. Regeneration of the Adsorbents

As methylene blue is particularly well soluble in ethanol and chloroform, a cheap and easy method to regenerate the adsorbents after adsorption was developed. A total of 150 mg of foam powders were washed with 150 mL of ethanol at different temperatures (ambient, 50 °C, and boiling) in a magnetically stirred beaker for 10 min. Afterward, the mixture was filtrated and the foam could be recovered. Furthermore, it is possible to separate the methylene blue from the ethanol at reduced pressure to repurpose the solvent.

The regenerated foams were further tested for their adsorption capacity and the results were compared with the first adsorption cycle.

### 2.7. Statistical Analysis

ANOVA and Tukey’s post hoc test were performed at a significance level of α = 0.05 using the IBM SPSS 25 statistics processor to determine significant differences between the adsorbents.

## 3. Results and Discussion

### 3.1. Pollutant Removal

Table 2 shows a summary of the removal capacities of the investigated polymer powders.

It was evident that the nature of the pollutant had a major influence in the remediation capacity of the powders: charged pollutants like the anionic surfactant SDS and even more of the cationic dye methylene blue were significantly adsorbed, while the neutral riboflavin was much less adsorbed. This observation confirms the finding of previous research when heavy-metal cations were absorbed [25]. It was also observed that (i) the higher initial molecular mass (acetone insoluble and tannin-lignin) and (ii) the highly cross-linked (formaldehyde) polymers had a positive effect on the absorption of the pollutants.

A positive effect of the lignin was observed when compared to the results for adsorption by tannin based polymers in the literature. Similar materials adsorbed roughly 45 mg methylene blue per gram adsorbent and 22 mg SDS per gram of the tannin polymer [21]. The results also exceeded the adsorption capacity of other agricultural by-products like hazelnut shells [43] or spruce wood [7]. However, as soon as the adsorbents are specifically designed for the removal of pollutants, the absorptions skyrocket (e.g., 200 mg/g for the simultaneous removal of methylene blue and oil by polymer membranes [10] or lignin derived activated carbons with 550 mg/g [44]).

The adsorption mechanism of SDS on flavonoids was studied by Liu and Guo where they stated that the more negatively charged B and C ring interacted with the SDS [45]. This finding and the well-known interaction between SDS and proteins can explain the removal of SDS from solution by tannin based polymers. A further aspect to be considered for understanding the adsorption mechanism for methylene blue is the natural electrostatic attraction between the cationic dye and the flavonoid units [9].

Acidified tannin gels were produced and their pollutant absorption capacity were investigated to understand the contribution of tannin in the absorption process. The results of pollutant absorption capacity measured with the same parameters are reported in Figure 2, which highlights that the adsorptions obtained for this powder were much higher than that registered for the polyphenolic-furanic polymers.

These high adsorptions suggest that the major contribution to the absorption process is given by the tannin part, while the furanic moieties of the polymers limit the establishment of secondary forces with the pollutants. These unexpected results further addressed the interest in the chemistry of the adsorbents, therefore a detailed chemical characterization of the powders was carried out.

### 3.2. Adsorbent Characterization

In Figure 3, the FT-IR spectra of the tannin foam, acidified tannin gel, and tannin powder are presented.

The spectra of the tannin gel presented a similar profile to the standard foam except for a decrease in the signals typically due to the furanic moieties like the carbonyl group at 1703 cm^−1^ and the signal at 795 cm^−1^, which were attributed to furanic C–H bending (out-of-plane) [46]. The tannin gel also presented a very different profile to the tannin powder and particularly, a higher absorption at 1605 and 1510 cm^−1^, which suggests increased aromatic contribution as well as a general increase in the region between 1350 and 1100 cm^−1^, which is dominated by C_arom_–O vibrations [47]. The band at 1190 cm^−1^, which was higher for the tannin gel than for the foam, and the tannin powder could be attributed to the interflavonoid vibrations [48].

The solid state ^13^C-NMR analysis of the adsorbing powders and of the tannin powder was performed and is reported in Figure 4.

Comparing the spectra of the foam with that of the gel, we observed that the bands attributed to the furanic part decreased in the tannin gel, in particular, the carbonyl signal due to γ-diketons resulting from the furanic ring opening at 210–190 [49] disappeared, the furan band at around 110 ppm decreased, and the methylene signals between furans at around 30 ppm decreased. Additionally, the signals due to the second polymerization step of furfuryl alcohol decreased, especially the ones at around 80 and 40 ppm (due to furanic ring opening and Diels–Alder rearrangements) [50]. Differences between the tannin gel and the tannin powder could also be highlighted through this technique. In particular, we observed that the region between 85 and 60 ppm dramatically decreased, together with the small peak at 92 ppm. These signals can be attributed to carbohydrates, confirming that in the acidified tannin gel in this fraction was removed (e.g., hydrolyzed and leached out). Furthermore, the signal at 130 ppm was significantly increased together with the shoulder band at 45 ppm, while the shoulder at 97 ppm disappeared. These results suggest that the unreacted C6 and C8 of the tannin were involved in establishing new interflavonoid bonds [51,52].

The two spectroscopic investigations highlighted that tannin gel has a similar chemistry to the tannin foam without furanic contribution and differs from the tannin powder because: (i) the carbohydrate fraction is removed, and (ii) the molecular mass is increased through the establishment of further interflavonoid bonds. The gel is then constituted by higher molecular mass flavonoids, no furanics, and no sugars, which explains its outstanding absorption activity.

### 3.3. Regeneration

These adsorbing materials have shown an interesting absorption capacity, especially for charged pollutants, therefore their regeneration was considered by washing them with ethanol. Table 3 shows the regeneration of the standard foam after the methylene blue adsorption test at 21, 50, and 79 °C (the boiling point of ethanol).

It can be stated that almost all of the absorbing sites were already free after washing with ethanol at room temperature. This means that the foams can be almost completely regenerated with a contained removal decrease, suggesting that these materials are suitable for several removal cycles.

## 4. Conclusions

The present study had the objective to understand the possibility of using tannin and tannin-lignin furanic powder as a pollutant adsorbent in wastewater treatments. We discovered that these materials have high affinities for cationic dyes and anionic surfactants, while low absorption was observed for the neutral molecules (riboflavin). The flavonoid backbone of the macromolecule is the moiety that positively contributes to the absorption because acidified tannin gels present much higher absorptions. On the other hand, the furanic part limits the pollutant adsorption because the number of reactive sites on the flavonoids is reduced by the presence of furanic moieties. Furthermore, the higher dimension of the starting building blocks (e.g., lignin containing) and the higher crosslinking degree (e.g., formaldehyde, acidified gel) contribute to increasing the absorption capacity due to the minor chance for the furanic part to hinder the absorption activity of the flavonoids. This finding was also confirmed through spectroscopic analysis of the acidified tannin gels, which highlighted the increase of interflavonoid linkages in the tannin. Notwithstanding the limiting contribution of the furanic part, we can determine that the tannin foams can be used as a pollutant absorbent at the end of their life, especially because the absorbing capacity can be almost completely recovered after simple ethanol bathing.

## Figures and Tables

**Figure 1 polymers-11-00911-f001:**
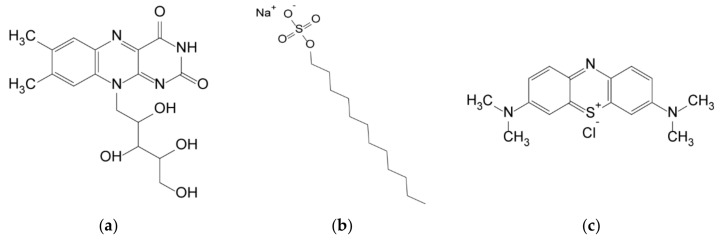
Chemical formula of the three pollutants tested: **(a)** riboflavin; **(b)** SDS; **(c)** methylene blue.

**Figure 2 polymers-11-00911-f002:**
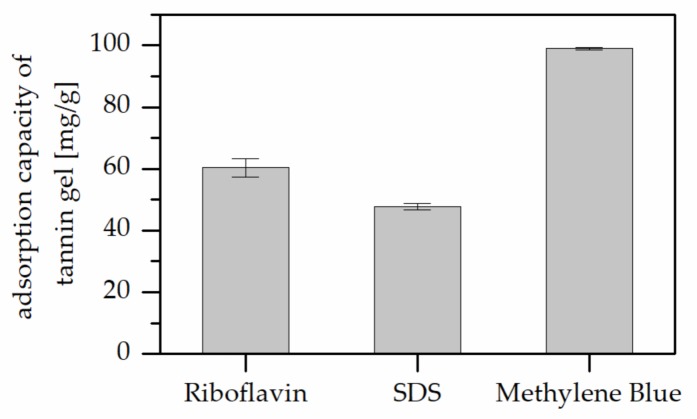
Adsorption capacity of the tannin gel against different pollutants.

**Figure 3 polymers-11-00911-f003:**
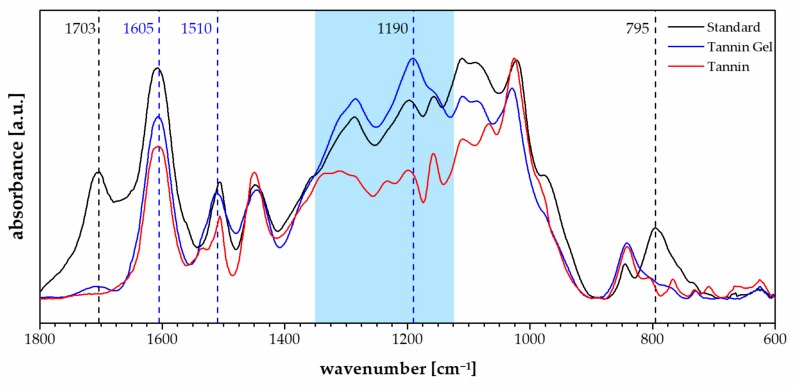
FT-IR spectra of the standard foam (black), the tannin acidified gel (blue), and the tannin extract (red).

**Figure 4 polymers-11-00911-f004:**
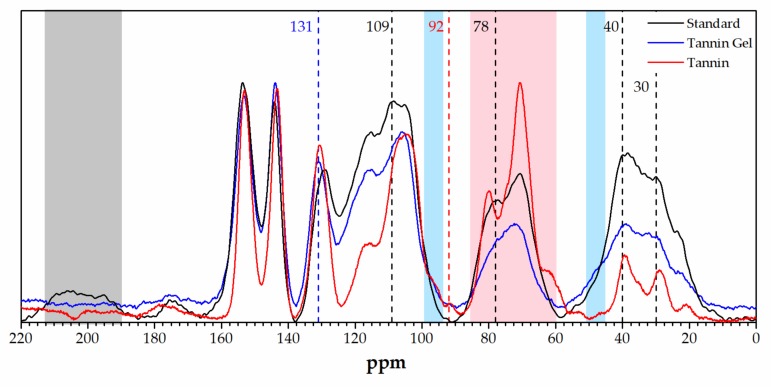
Solid-state ^13^C-NMR spectra of the standard foam (black), the tannin acidified gel (blue), and the tannin extract (red).

**Table 1 polymers-11-00911-t001:** Formulations used for foam production.

Foam	Tannin [%]	FOH [%]	DEE [%]	H_2_O [%]	H_2_SO_4_ [%]	CH_2_O/BL [%]	Temperature [°C]
Standard	41.9	25.9	5.6	8.4	18.2	-	90
Acetone soluble	41.9	25.9	5.6	8.4	18.2	-	90
Acetone insoluble	41.9	25.9	5.6	8.4	18.2	-	90
Tannin-Lignin	29.6	20.7	3.3	-	16.8	29.6	120
Formaldehyde	44.3	27.5	5.9	-	13.4	8.9	45

**Table 2 polymers-11-00911-t002:** Adsorption capacities [mg/g] of the foams against different pollutants.

Foam	Riboflavin ^1^	SDS ^1^	Methylene blue ^1^
Standard	3.8 ± 0.81 ^a,b^	24.1 ± 0.31 ^a^	45.4 ± 0.31 ^a^
Acetone soluble	4.8 ± 0.88 ^a,b,c^	20.3 ± 0.34 ^a^	33.2 ± 1.29 ^b^
Acetone insoluble	7.6 ± 0.28 ^c^	32.7 ± 2.05 ^b^	63.3 ± 2.33 ^c^
Tannin-Lignin	6.0 ± 1.41 ^b,c^	32.8 ± 2.46 ^c^	73.8 ± 2.40 ^d^
Formaldehyde	2.1 ± 1.48 ^a^	32.6 ± 2.05 ^b^	56.9 ± 1.92 ^e^

^1^ Same letters in a column suggest no significant difference at α.

**Table 3 polymers-11-00911-t003:** Adsorption capacity of the standard foam washed in ethanol at different temperatures.

Foam Condition	Removal Capacity [mg/g]	Compared to Initial Removal [%]
Fresh	45.38 ± 0.31	-
Washed at 21 °C	44.03 ± 0.21	97.6
Washed at 50 °C	44.96 ± 0.58	99.2
Washed at 79 °C	44.77 ± 0.48	98.7

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
