# Peer review of "Pollutant Absorption as a Possible End-Of-Life Solution for Polyphenolic Polymers"

_polymers, 2019, doi:10.3390/polym11050911_

Round 1

Reviewer 1 Report

Manuscript titled ‘Pollutant absorption as end-life solution for tannin foams’ by Sepperer et al. describes tannin-lignin based foams as efficient adsorbent for the cationic or charged adsorbate. This work looks too preliminary as the work reported has not been designed as a full scientific article on this subject as available in literature. Though the results reported look good alongwith good material characterization, yet this work has major limitations which need to be improved in the revised manuscript.

1.      Novelty of the work has not be explicitly brought out. Authors abruptly ended the introduction part without explaining the objective of the work in some details and making a comparative statement citing literature?

2.      High temperature of 120 oC has been used for tannin-lignin foam thus vitiating the energy gains cf. 45 oC for the FA  

3.      The adsorption scheme looks too limited. How and why the authors choose particular adsorption conditions?

4.      Choosing 20 ppm adsorbate solution is too low for real samples? Similarly, stirring the adsorption experimental set-up for 48h under the dark is too long time period for real conditions.

5.      Choice of the pollutants chosen needs to be explained.

6.      Why authors chose ethanol as eluent is not explained. What are the specific conditions used to generate data presented as Figure 2.

7.      Authors conclusion ‘The flavonoid backbone of the foam is the moiety that positively contributes to the absorption because acidified tannin gels present much higher absorptions while the furanic part have a negative contribution.’ This statement needs to be revisited by the authors on two counts. One, they have not reported surface charge of the foam which from the result looks –ve. Two, ‘a negative contribution’ looks out of place. It may not contribute to adsorb cationic dye, but it should not also play a role to reduce adsorption?

8.      Citations on adsorption of these three pollutants alongwith comparative statements under the Results and Discussion part needs to be added to enrich this manuscript.

Author Response

Reviewer 1:

Manuscript titled ‘Pollutant absorption as end-life solution for tannin foams’ by Sepperer et al. describes tannin-lignin based foams as efficient adsorbent for the cationic or charged adsorbate. This work looks too preliminary as the work reported has not been designed as a full scientific article on this subject as available in literature. Though the results reported look good along with good material characterization, yet this work has major limitations which need to be improved in the revised manuscript.

1.      Novelty of the work has not be explicitly brought out. Authors abruptly ended the introduction part without explaining the objective of the work in some details and making a comparative statement citing literature?

We thank the reviewer and we agree with his/her observation. After carefully reading the introduction we have found that the red-line of the introduction was not easy to follow. Therefore we have restructured and extended the introduction with examples of work done in the field of wastewater treatment. Novelty, in the meaning of an alternative use for tannin foams after their initial purpose, was added at the end of the introduction. In order to further embed the paper in its application field we added comparison to the literature also in the Results and discussion part.

2.      High temperature of 120 oC has been used for tannin-lignin foam thus vitiating the energy gains cf. 45 oC for the FA.

Thank you for the cue. Tannin-formaldehyde foams can expand at low temperature because the polymer produced with formaldehyde is tougher. The tannin-lignin foams (formaldehyde-free) require elevated temperature, because the reaction between tannin, lignin and furfuryl alcohol need higher activation energy.

3.      The adsorption scheme looks too limited. How and why the authors choose particular adsorption conditions?

This is a good remark. Our study had the objective of finding further application for our polyphenolic polymers at the end of their life. Therefore, we have tested only three pollutants, in one concentration. Now that the results are very promising we will consider to optimize this study by extending the investigated parameters.

4.      Choosing 20 ppm adsorbate solution is too low for real samples? Similarly, stirring the adsorption experimental set-up for 48h under the dark is too long time period for real conditions.

We agree with the reviewer that 20 ppm is low for a real sample. However, the relation of adsorbent to solution has shown good reproducibility at our instruments, while higher initial concentration of methylene blue made it difficult to detect the differences.

Additional we chose 48h to ensure complete saturation of our adsorbents. The darkness was necessary to avoid the effects of photo-degradation.

5.      Choice of the pollutants chosen needs to be explained.

We agree with the reviewer. We explained the selection by considering the context in the introduction and into details in the experimental section.

6.      Why authors chose ethanol as eluent is not explained. What are the specific conditions used to generate data presented as Figure 2.

The selection of the regenerating eluent was due to the fact that methylene blue is easily soluble n ethanol and then the recovery of ethanol is manageable by simple reduced pressure evaporation. This detail was added in the experimental part.

b) clarified the conditions for obtaining data in graph 2.

The same absorption conditions were applied and this was further highlighted in the text introducing Figure 2.

7.      Authors conclusion ‘The flavonoid backbone of the foam is the moiety that positively contributes to the absorption because acidified tannin gels present much higher absorptions while the furanic part have a negative contribution.’ This statement needs to be revisited by the authors on two counts. One, they have not reported surface charge of the foam which from the result looks –ve. Two, ‘a negative contribution’ looks out of place. It may not contribute to adsorb cationic dye, but it should not also play a role to reduce adsorption?

Thank you for the comment. We believe that the furanic moieties may cover some flavonoid part by giving a negative contribution, however this cannot b easily quantified and therefore we stated limiting contribution. This statement have been corrected all over the text.

8.      Citations on adsorption of these three pollutants along with comparative statements under the Results and Discussion part needs to be added to enrich this manuscript.

As already previously stated, we have extended the introduction, clarified the selections in the experimental part and finally also added a new paragraph in the discussion part for better embed this paper in the state of the art of the field.

Reviewer 2 Report

The manuscript “Pollutant absorption as end-life solution for tannin foams” was well performed.

1)      The title can not well have described the subject.

The title highlight tannin foams, but the paper not showed any characterization on foams (such as microstructure, pore properties, BET test). All the works are based on the powder rather than porous monoliths or components. Why authors not do the absorption test based on foams?

2)      The paper can be published as a letter or communication. As a regular article, the results and discussion is not insufficient. The parameter optimization for the adsorption (pH, contact time, adsorbent dosages, etc.) should be added. The adsorption mechanism should be discussed. Pollutant removal capacity should be compared with previous works or relevant materials.

3)      Besides the pore structure properties, the foaming mechanism is not discussed.

4)      The details for SDS adsorption should be added. As this manuscript in not a letter. Only listed the reference is not insufficient.

Author Response

Reviewer 2:

The manuscript “Pollutant absorption as end-life solution for tannin foams” was well performed.

1)      The title cannot well have described the subject.The title highlight tannin foams, but the paper not showed any characterization on foams (such as microstructure, pore properties, BET test). All the works are based on the powder rather than porous monoliths or components. Why authors not do the absorption test based on foams?

This is a very nice point. The reviewer is correct. We have changed the title of the manuscript to “Pollutant absorption as possible end-of-life solution for polyphenolic polymers”

2)      The paper can be published as a letter or communication. As a regular article, the results and discussion is not insufficient. The parameter optimization for the adsorption (pH, contact time, adsorbent dosages, etc.) should be added. The adsorption mechanism should be discussed. Pollutant removal capacity should be compared with previous works or relevant materials.

We thank the reviewer for the remarks. Our study had the objective of finding further application for our polyphenolic polymers at the end of their life. Therefore, we have tested only three pollutants, in one concentration. Now that the results are very promising we will consider to optimize this study by extending the investigated parameters.

About including this work in a specific context we have added a comparison to the literature in the result & discussion part. Adsorption mechanism was discussed by explaining the strong absorptions of the acidified tannin gels.

About the article length, we agree that our paper is not the longest, but at the same time we think that the data collected and the tests done can be hardly compressed in a short communication.

3)      Besides the pore structure properties, the foaming mechanism is not discussed.

As the research focuses more on the possible end-of-life scenario than the foams themselves, literature references for the synthesis have been added.

4)      The details for SDS adsorption should be added. As this manuscript in not a letter. Only listed the reference is not insufficient.

Thank you for your advice. In the discussion part of the paper we have added the explanation for the SDS absorption that we found in the literature. The introduction was also extended and further 17 papers have been added.

Round 2

Reviewer 1 Report

Thanks for addressing all the concerns I had on the original manuscript.

Reviewer 2 Report

after minor revision, 

There are two Table 1. Please check!.

Now this paper Pollutant absorption as possible end-of-life solution for polyphenolic polymers can be accepted

as short communication.

.